# Near-Infrared Spectroscopy as a Tool for the Traceability Control of High-Quality Iberian Dry-Cured Meat Products

**DOI:** 10.3390/foods14030432

**Published:** 2025-01-28

**Authors:** Alberto Ortiz, Lucía León, María Rosario Ramírez, David Tejerina

**Affiliations:** 1Meat Quality Area, Centre of Scientific and Technological Research of Extremadura (CICYTEX-La Orden), Junta de Extremadura, Ctra A-V, Km372, 06187 Badajoz, Spain; alberto.ortiz@juntaex.es (A.O.); david.tejerina@juntaex.es (D.T.); 2Technological Institute of Food and Agriculture (INTAEX), Centre of Scientific and Technological Research of Extremadura (CICYTEX), Avda. Adolfo Suárez s/n, 06007 Badajoz, Spain; mariarosario.ramirez@juntaex.es

**Keywords:** Iberian dry-cured product authentication, high hydrostatic pressure, traceability, pre-packaged sliced format, non-destructive, room temperature

## Abstract

Near-infrared spectroscopy (NIRS) was evaluated to trace the high hydrostatic pressure (HHP) processing and preservation temperature (4 °C vs. 20 °C) over the course of a long term in vacuum-packaged Iberian dry-cured *tenderloin* (*Iliopsoas et psoas minor*). Spectra were obtained from a total of 298 samples, without opening the package, using a handheld MicroNIR^TM^ 1700 OnSite-W microspectrophotometer (908.1 nm–1676.2 nm) (VIAVI Solutions Inc., United States). The discriminant models were developed by means of partial least squares-discriminant analysis (PLS-DA). The models obtained were capable of correctly classifying more than 60% of the samples according to their HHP processing, while almost 100% of the samples were correctly classified according to the temperature at which the samples were preserved. Thus, NIRS could help to support the traceability of treatments that represent a high added value to the product, such as HHP in premium Iberian dry-cured products.

## 1. Introduction

The Iberian breed is an autochthonous breed of pig traditionally reared in the southwest of Spain, providing high-quality products, especially those from animals reared under *Montanera*. The traditional and most widely accepted way of consuming Iberian products is dry-cured products, with the most common and well-known pieces being the hams and shoulders. However, recently, new commercial cuts for fresh consumption, such as the *tenderloin* (*Iliopsoas et psoas minor*), the *presa* (*Serratus ventralis*), and the *pluma* (*Spinalis dorsi*), are currently hitting the market, as a strategy by the Iberian sector for innovation and competitiveness [1]. Among these, the production and commercialization of tenderloin as a dry-cured product may pose a technological challenge and give the product an advantage in terms of innovation, since until now it has only been commercialized for fresh consumption [2].

On the other hand, the trend in the selling format of dry-cured products is moving towards a ready-to-eat format, i.e., sliced and packaged. Meanwhile, the trend in terms of preservation during commercialization is towards room temperature [3], in response to the current demand from industry and supply chains to promote competitive strategies and simplify logistics operations, as well as the need to align with current energy-saving and efficiency policies. This, however, raises new concerns in relation to quality loss and the assurance of microbiological safety, which may be affected by slicing and packaging operations, and promoted by subsequent storage at room temperature, as demonstrated [3] for vacuum-packed sliced Iberian dry-cured ham. Of particular concern in this regard is the pathogenic bacteria *Listeria monocytogenes*, as many countries apply a “zero tolerance” policy [4].

High hydrostatic pressure (HHP) processing is a non-thermal treatment that involves applying hydrostatic pressures to the product, which may be between 400 and 600 MPa, to ensure microbiological stability. The success of this technique in destroying pathogenic and spoilage microorganisms results from the damage high pressure causes to cell walls. The application of HHP to dry-cured products, especially those considered of high quality and added value, such as those from Iberian breed pigs, has been extensively explored in recent years [3,5,6,7], proving its potential to reduce microbiological counts and to ensure the destruction of *Listeria monocytogenes* [8]. Nevertheless, United States Department of Agriculture [5] recently concluded that the initial microbial decline induced by HHP could be softened during storage, especially when storage is conducted under room-temperature conditions.

In view of the relevance that both factors—HHP treatment and preservation temperature—may have on the stability of dry-cured meat products, it would be of great interest to have quick tools that would make it possible to trace in situ if a given product has been subjected to a pressurization treatment or under which storage conditions it has been preserved. In this way, the added value of the pressurization treatment or of keeping the product under refrigerated conditions compared to room temperature would be safeguarded. Consequently, it would help the industry and supply chains to optimize the time, labor, and economic resources needed to carry out the required control for dry-cured products under these selling formats. Moreover, considering the high value of this type of product, it is important that this traceability control can be performed in a non-destructive manner. In this sense, one of the pig industry’s most used and demanded technologies is the near-infrared spectroscopic technology (NIRS), which makes it possible to obtain information and make real-time decisions in a non-destructive manner, therefore providing a great advantage compared to conventional analysis methods, given the high value of Iberian pig products. Scientific literature has consolidated NIRS technology as a useful tool to ensure quality control, traceability, and authenticity in meat products. Thus, [9] demonstrated the possibility of classifying live animals, carcasses, fresh meat, and subcutaneous backfat according to the animal’s feeding regime (acorn vs. Feed) by means of NIRS. Later, [10] reported the ability of this technology to discriminate Iberian fresh loins from those that had previously been frozen-thawed. In addition, the use of the NIRS technology could also be feasible for the individual control of dry-cured products in sliced packed format, therefore protecting and assuring their traceability, providing additional value to the product. Specifically, in this regard, previous research studies have demonstrated how NIRS technology may be used for the purpose of assigning the official commercial category defined by the Iberian Quality Standard [11] in Iberian dry-cured loin [12], assuring their origin and therefore making sense of the labeling system. The latter research study also showed the potential of this technology for the traceability of product preservation conditions (discrimination of conventional storage under refrigerated conditions versus storage at a temperature of 20 °C). Recently, NIRS technology has been applied to dry-cured products under sliced and packaged formats of other pig breeds than the Iberian breed, but equally considered of high eating quality, such as the Duroc breed. Specifically, León et al. [13]. concluded the ability to classify Duroc dry-cured ham according to preservation conditions as well as its storage time prediction, considering two packaging conditions: vacuum and modified atmosphere. However, the scientific literature is nonexistent in relation to the quality control of the Iberian dry-cured *tenderloin*. Furthermore, to the best of our knowledge, the capability of NIRS technology to trace HHP treatment has not yet been addressed.

In this context, this research study was conducted with the aim of evaluating the potential of NIRS technology—using a portable MicroNIR^TM^ 1700 OnSite-W microspectrophotometer—to trace HHP treatment and storage temperature (4 °C vs. 20 °C) on vacuum-packaged Iberian dry-cured *tenderloin* in a non-destructive way, i.e., without opening the package.

## 2. Materials and Methodology

### 2.1. Iberian Dry-Cured Tenderloin Samples

A total of 80 pieces of *tenderloin* (*Iliopsoas et psoas minor*), commercially categorized as *Black* label according to the current official Spanish Iberian quality standard [11], acquired in Señorío de Montanera, Badajoz, Spain, were randomly selected and used to carry out this research. The pieces were purchased from a specialized Iberian meat industry; Señorío de Montanera S.L., Badajoz, Spain. The technological process of curing was undertaken in the aforementioned industry and in accordance with standard industry practice as follows. The commercial cuts (*Iliopsoas et psoas minor*) were seasoned with a mixture of salt (2.50%), lard (1.70%), spices (<0.2%), and preservatives and authorized stabilizers (E-320 < 0.1%, E-321 < 0.1%, dextrose < 0.1%, E-252 < 0.1%, E-250 < 0.1%, and E-301 < 0.1%). They were then stored under refrigerated conditions (4 °C ± 2) for 96 h in the dark to allow the seasoning mixture to penetrate the meat. They were then stuffed into 60 mm diameter collagen casings and subjected to the technological curing process in which the relative humidity gradually increased from 79% to 85% while the temperature was adjusted from 4 °C to 14 °C over the course of 50 days.

Next, the pieces of *tenderloin* were sliced into 2-mm-thick slices under commercial conditions and arranged in 100 g-vacuum package formats using a laminated film (oxygen permeability, 9.3 cm^3^ O^2^/m^2^/24 h at 4 °C), by means of an EGARVAC^®^ (Vacarisses, Barcelona) packaging unit, obtaining a total of 298 packages. Each package contained slices from various pieces of tenderloin to allow for sufficient representativeness of the potential variability existing among pieces in each package, which was the experimental unit.

The packages were divided into the various experimental groups: (1) non-pressurized vacuum packages and stored at refrigeration conditions (4 °C ± 2); (2) non-pressurized vacuum packages and stored at room temperature (20 °C ± 2); (3) vacuum packages HHP-treated and stored in refrigeration conditions; and (4) vacuum packages HHP-treated and stored at room temperature (Table 1). The samples under room temperature were stored in a climate-controlled warehouse. Every day, the temperature was measured twice and never varied more than 1.5 °C.

The HHP treatment was of 600 MPa for 8 min in a semi-industrial hydrostatic pressure unit with 55 L of capacity (6000/55, Hiperbaric, S.A., Burgos, Spain). The initial water temperature inside the vessel was 15 °C. The time to reach 600 MPa was 230 s. The depressurization time was instantaneous. The conditions of the treatment were chosen to ensure the inactivation of pathogens, such as *Listeria* spp. [7].

Spectra were taken from samples sampled at the beginning of the experiment (T0), and after 4 (T4) and 8 (T8) months of storage (Table 1). The spectra taken at the beginning of the experiment (T0) were considered for the discrimination of the pressurization treatment, but not for the discrimination of the preservation temperature, since obviously, at T0, this factor could not have had any effect on the samples.

### 2.2. NIR Spectra Acquisition

The NIR spectra were acquired using a MicroNIR^TM^ 1700 OnSite-W microspectrophotometer (VIAVI Solutions, Inc., San Jose, CA, USA), in the spectral range between 908.1 nm and 1676.2 nm, and spectra were recorded at intervals of 6 nm (Figure 1). The integration time was 10 ms, and the spectral resolution <1.25% of the center wavelength. This device was chosen for its portability and hence its appropriateness for utilization within the industrial and meat supply chain. The reference for the 0% reference value (known as the dark current scan) was acquired by leaving the tungsten Lampson with an empty support in a stationary position within the room. Conversely, since the aim of this study was to generate predictive models without opening the packages, i.e., in a non-destructive way, the 100% reference value was performed using the NIR reflectance standard (spectral on ceramic tile of politetrafluoroetilen (~99%) with a 99% diffuse reflectance) covered with the same laminated film material utilized for packaging the product. A single spectrum, which was an average of 500 scans, was obtained for each sample using a direct contact instrument-package approach. The instrument was moved in a zigzag pattern across the surface of the package to increase the sampled area and minimize sampling errors. The instrument handling and data recording were performed using the MicroNir Pro v2.2 software (VIAVI Solutions, Inc., San Jose, CA, USA).

### 2.3. Chemometric Analysis

After NIR spectra acquisition, spectral data –(log1/R), where R is the reflectance, were exported into the Unscrambler X vs. 10.5 (CAMO^®^, Trondheim, Norway) for the data processing and modeling.

Out of the total number of samples, about 70% were selected and used for the construction of the classification models (the calibration set), while the rest (30%) were subsequently used for the validation of the models (external validation set). It was conducted by manual and random selection according to the experimental design, ensuring the representation in each subset of samples from the various experimental groups to maximize variability in both calibration and validation sample tests (Table 1).

Data modeling was performed to reduce the dimensionality and to extract useful information from the spectral data by transforming it into a new dataset of more manageable size. The development of the discriminatory models for the pressurization treatment and the preservation temperature of Iberian dry-cured *tenderloin* were performed on the set of calibration samples using the partial least squares algorithm discriminant analysis (PLS-DA). This is a supervised classification method that correlates spectral variations (X variables) with the defined classes (Y variables, i.e., HHP-treated samples and control ones in the first place, and samples preserved under refrigeration conditions and those preserved at room temperature over the course of 8 months, in second place). So, according to this approach, Y variables act as dummy variables [14]. Thus, a value of 1 is assigned to the samples belonging to the target category to be discriminated from the rest of the categories that assume a value of 0. This therefore allows the use of the partial least squares algorithm in qualitative cases. The PLS-DA method seeks to optimize the latent variables by maximizing the covariance between X and Y. This is achieved by reducing the data to scores and the loading matrix, aiming to find the most optimal representation of the information. The selection of latent variables is based on the lowest cross-validation error, determined using the leave-one-out method.

The models were developed using the original spectral data (absorbance) as well as after certain spectral pre-treatments, individually or in combination, which were applied in the full spectra range, i.e., from 908.1 nm to 1676.2 nm. Thus, the multiplicative scatter correction (MSC) and the standard normal variable (SNV), the latter in combination with the de-trending (DE), were used to correct dispersion phenomena. Furthermore, in the best of the pre-treatments used to correct dispersion phenomena, mathematical derivatization treatments were also applied using the Savitzky-Golay (SG) approach. In particular, two derivatives were examined. The first derivative was obtained using a symmetric kernel with 4 smoothing points on both the left and right sides, and a first polynomial order (1,4,4,1). The second derivative was obtained using a symmetric kernel with 5 smoothing points on both the left and right sides, and a second polynomial order (2,5,5,2).

During the construction of the models, any outliers that were identified through spectra plotting performed by principal component analysis (PCA) were removed. The elimination criteria used was a leverage (H) higher than 3 times the average leverage. This was done to ensure that the models were not influenced by extreme or influential observations [15]:

H = 1/n + (number of principal components/n), in which “n” refers to the number of samples.

### 2.4. Evaluation of Qualitative Prediction Models

The models were evaluated by means of the determination of the coefficient after cross-validation (1-VR) and root mean square error after cross-validation (RMSECV). The best model was chosen based on the highest value of 1-VR, and the lowest RMSECV (Appendix A). Additionally, the best-fitting models were validated by means of an independent sample set—the external validation set—to ensure the reliability of the models. The external validation of the models was assessed by the Sensitivity (SE), Specificity (SP), and Classification accuracy statistics [16].

### 2.5. Reference Analysis and Statistical Analysis

The antioxidants compounds, α- and γ-tocopherol, were determined by the method proposed by [17], as widely described in [18].

The main groups of fatty acids (saturated, mono-, and polyunsaturated fatty acids; SFA, MUFA, and PUFA, respectively) were determined from the fat extracted according to [19]. The gas chromatograph conditions are specified in [18].

Regarding the oxidative status, lipid oxidation was measured using 2-thiobarbituric acid (TBA) [20] and expressed as µg malondialdehyde (MDA) g^−1^ of the sample. Protein oxidation was determined by measuring the carbonyl groups formed during incubation with 2,4-dinitrophenylhydrazine (DNPH) in 2N HCl [21]. The results were expressed as nmol carbonyls mg protein-1.

The statistical analysis of the data consisted of two multivariate analyses of variance (one-way ANOVA) using the software package SPSS.PC + v. 20.0 to evaluate the effect of both factors (the HHP treatment and the storage temperature over the course of 8 months) on antioxidant compounds, the main fatty acids groups, and the oxidative status of Iberian dry-cured *tenderloin*, establishing a significance level of *p* = 0.05. The results were expressed as mean values ± standard error.

The study of the effect of HHP treatment at the physical-chemical level was carried out on a total of 100 samples (50 HHP-treated and 50 untreated or control). Both groups included samples from the various storage conditions (4 °C and 20 °C). Similarly, for the study of the influence of storage temperature on physical-chemical parameters, a total of 80 samples were used; 40 stored under refrigerated conditions and 40 under room temperature, with HHP-treated samples and control ones included in each of these groups.

## 3. Results

### 3.1. Spectra Information

Figure 2 shows the mean spectra (absorbance) of the total samples grouped according to the HHP treatment (Figure 2A) or according to the temperature at which they were preserved (Figure 2B). Regarding spectral features, a similar pattern may be observed in these mean spectra, regardless of the pressurization treatment and/or temperature to which the samples were subjected during the subsequent storage and therefore displaying the same peaks and valleys across the entire spectra range (from 908 to 1676 nm). Nevertheless, differences in absorbance intensity were observed between spectra from samples that were subjected to the HHP treatment and the control ones. These differences occurred especially in the main peaks, i.e., around 950, 1200, and 1440 nm. Likewise, the temperature at which the samples were stored resulted in differences in the magnitudes of absorption at the aforementioned wavelengths.

Additionally, in order to obtain a more accurate interpretation of the areas with more useful information for the discrimination of HHP treatment and storage temperature, the loadings and weighted regression coefficients in the projection of the best-fitting PLS-DA model of each model were calculated (Figure 3A,B).

### 3.2. Development and Validation Models for Discrimination of the HHP Treatment

The statistics obtained from the cross-validation and external validation of the best-fit qualitative model for discriminating HHP-treated versus untreated Iberian dry-cured *tenderloin* samples are shown in Table 2, while Appendix A shows the calibration and cross-validation statistical results for all the pre-treatments used. The best-fitting prediction models were obtained after the combination of both SNV + DE and the SG 1,4,4,1 derivate. The model yielded a value of 0.78 for 1-VR and 0.233 for RMSECV. As far as external validation results are concerned, the SE and SP exceeded 60%, resulting in a similar value in accuracy.

### 3.3. Development and Validation Models for Discrimination of the Preservation Temperature

The best model to predict the temperature at which the product was preserved (4 °C vs. 20 °C) was obtained after applying SNV + DE in combination with the SG 1,4,4,1 derivate (Table 3). This offered high classificatory capacity, even higher than that observed for HHP treatment discrimination, yielding a 1-VR of 0.94, while the RMSECV was 0.122. The predictive capacity was confirmed after validation of the model with the external validation set, resulting in values of 97.22 for SE, 100% for SP, and 98% for accuracy. The results deriving from the use of several pre-treatments and their combination for discriminating the storage temperature are presented in Appendix A.

### 3.4. Effect of the HHP Processing and Preservation Temperature on the α- and γ-Tocopherol, Fatty Acid Profile, and Oxidative Status

The effect of the HHP processing and preservation temperature on the tocopherol content, fatty acid profile, and oxidative status were added to this study since the possibility of tracing both factors by means of NIRS technology may depend on these parameters. As far as antioxidants content is concerned, no differences were observed due to the HHP treatment (Table 4). Nevertheless, both alpha- and gamma-tocopherol compounds were significantly affected at the end of the storage period in the samples stored at room temperature (Table 5). In the case of lipid profile, a significant increase in SFA, as well as a decrease in MUFA and PUFA groups, was observed in samples preserved at 20 °C compared to those preserved under refrigerated conditions (Table 5), while pressurization did not introduce relevant changes in those groups (Table 4). Regarding the oxidative status, higher lipid and protein oxidation was observed in the HHP-treated samples, given by a greater value of MDA and carbonyls, respectively (Table 4). On the other hand, the storage of the product at 20 °C led to a higher lipid oxidation ratio compared to storage under conventional refrigeration conditions, but variations in carbonyls values according to the storage temperature were not significant (Table 5).

## 4. Discussion

The PLS-DA results obtained in this study were in line with previous studies concerning the application of NIRS technology for qualitative purposes. Thus, Horcada et al. [9] reported SE results between 50% and 85% and SP ranging from 82% to 94% after validating the models for the classification of Iberian fresh meat (*psoas major*) according to the various official commercial categories compiled by the Iberian Quality Standard [11], using a handheld spectrophotometer (MicroPhazir 1624, Polychromix, Inc., Wilminton, MA, USA) and with a spectral range between 1600 and 2400 nm. Later, Cáceres-Nevado et al. [10] obtained an accuracy of around 100% for the distinction of Iberian fresh meat from frozen-thawed meat by means of the same instrument and spectral range used in this study. Regarding dry-cured products, recently, Tejerina et al. [12] reported SE and SP values of between 43.75 and 74.19% for the prediction of the official commercial category of pre-sliced and modified atmosphere packaged (MAP) Iberian dry-cured loin. In this case, the instrument used was a LabSpec 5000 (ASD Inc., Falls Church, VA, USA) fitted with an ASD fiber-optic Contact Probe^®^ and the spectral range considered was from 1000 to 1800 nm. Continuing along the line of sliced and packaged dry-cured products, recently, Ortiz et al. [18] reported successful classification models for the case of vacuum- and modified atmosphere-packed (MAP) Duroc dry-cured ham. The models, obtained by means of PLS-DA and from spectral data taken with the same NIR device, correctly classified the 100.00% samples under vacuum packaging and more than 92.00% in the case of samples under MAP, belonging to the validation set, according to the storage temperature. Furthermore, good accuracy was obtained for the assignments into storage times, with values of between 92.31% and 100.00% for samples under vacuum and between 91.00% and 97.00% for those under MAP, in both cases after validation. The ability to trace the HHP treatment as well as the preservation temperature may be attributed to absorbance differences. Indeed, absorption bands observed in the NIR spectra are characterized by specific functional chemical groups, representing a combination of stretching, bending, and other molecular motions that are present in various molecules. As a result, the overall NIR spectra may provide valuable information about the chemical composition and molecular structure of the sample and, therefore, may serve as a unique fingerprint. Specifically, absorption bands between 920–930 nm and 1200 nm have been previously related to the C-H third and second stretching overtone, respectively [22,23], which are all characteristics of the fatty acid and tocopherols molecules. Therefore, the differences in absorbance intensity between the raw mean spectra of Iberian dry-cured *tenderloin* according to the HHP treatment or preservation temperature could be the result of physicochemical and/or nutritional changes in the product because of these factors, and more specifically due to the higher ratio of lipid oxidation in the HHP-treated samples. The pro-oxidative effect of the HHP treatment in meat products has been previously described, especially when the threshold between 300 and 400 MPa is exceeded [24], which has been attributed to metal catalysts (probably iron) released from complexes during pressure treatment [25]. Specifically, recent studies on dry-cured meat products point to an increase in the sensitivity of the HHP-treated product to lipid oxidation during subsequent storage, rather than an immediate pro-oxidant effect. In line with this, Rivas-Cañedo et al. [26] observed higher MDA values in HHP-treated Serrano dry-cured hams (600 MPa, 6 min) after refrigerated conditions storage for 5 months compared to the ones not treated. In the same line, the lack of an immediate pro-oxidant effect was also identified in Iberian sausages after 600 MPa for 8 min [7,8] and for the case of the Iberian loin [5]. On the other hand, the storage of the product at 20 °C led to lower antioxidant values—tocopherols, changes in the lipid profile, and a higher lipid oxidation ratio compared to storage under conventional refrigeration conditions (Table 5). This could be explained by the fact that the rate of lipid oxidation generally increases with temperature [6,27]. Lipid oxidation processes, in turn, could explain the lower value of tocopherols observed in the samples stored at 20 °C, since tocopherols act as primary or chain-breaking antioxidants by converting lipid radicals into more stable products [28]. Similarly, the lower proportion of unsaturation of the fatty acid profile of these samples (lower MUFA and PUFA value) could result of lipid oxidation processes, which mainly affect them [29]. Lastly, the absorption bands around 1440 nm in the NIR spectra correspond to the N-H stretch first overtone associated with the CONH_2_ group found in peptide bonds of proteins. Therefore, these spectral features could provide important information about the characteristics of protein structures. Therefore, the differences in absorbance intensity of the mean spectra observed in this region could reflect the differences in protein oxidation associated with the higher protein oxidation observed in the samples subjected to the pressurization treatment with respect to the control samples (3.91 vs. 3.49 nmol carbonyls/mg protein, respectively, *p* = 0.003). The effect of HHP on protein oxidation has not been as comprehensively studied as in the case of the lipid oxidation, and the scientific literature provides conflicting results, possibly because it depends on aspects such as the type of meat or HHP treatment conditions [24]. In any case, our results are in agreement with those reported by Cava et al. [8] for the case of Iberian dry-cured loin, in which pressurization significantly increased carbonyls formation after 60 days of storage. These results are in accordance with the loadings and weighted regression coefficients in the projection of the best-fitting PLS-DA models (Figure 3A,B). From these plots, it is possible to draw the most important regions used by the PLS-DA model to discriminate samples according to HHP treatment or storage conditions, especially looking at the loading values which provide more quantitative information about the contribution of each region of the NIR spectrum to the qualitative model. Thus, the most important regions to discriminate HHP-treated samples from control ones were those around 1200 nm and 1440 nm, while the wavelengths that were more useful to classify the samples according to the storage temperature were between 1160 and 1260 nm. In general, there is consensus in the scientific literature regarding the relevance of these regions of the NIR spectrum for classification purposes in other similar products. Specifically, Pérez-Marín et al. [30] ascribed the classificatory ability of Iberian pig carcasses according to the animal’s diet during their last fattening phase (acorn vs. commercial feedstuff) to regions of the C-H bond absorption bands. Similarly, Tejerina et al. [12] proposed that the variations in absorbance in spectrum signature in the regions of the C-H bond absorption bands may be the basis for the classification of the pre-sliced and MAP Iberian dry-cured loin. Later, Cáceres-Nevado et al. [10] described regions between 930–960, 1130–140, 1200–1230, and 1350–1450 nm as informative for the differentiation between fresh and frozen-thawed Iberian loin. These authors attributed these bands to chemical changes resulting from freezing, including lipid hydrolysis and protein in solubilization, among others. Limitations that might have an influence on the obtained results include the composition of the surrounding plastic material, as well as the conditions of the HHP treatment and preservation temperature, since these have an impact on the physical-chemical parameters of the product and therefore on the differences at the spectral level.

## 5. Conclusions

The results obtained in this study suggest the possibility of achieving traceability control of high hydrostatic pressure treatment, since the models obtained were capable of correctly classifying more than 60% of the samples according to their HHP processing. Similarly, reliable classification models were obtained for the traceability of the temperature to which Iberian dry-cured *tenderloin* was preserved throughout long-term storage (4 °C vs. 20 °C). In this case, the models correctly classified almost 100% of the samples according to the temperature at which they were preserved.

The potential of NIRS technology for the discrimination of both factors in dry-cured products in conjunction with the acquisition of the spectrum in a non-destructive way, i.e., without even opening the package, could make it feasible for NIRS technology to be applied by meat processing plants and other actors of the distribution supply chain and therefore to protect the added value that these products may have after HHP treatment or if they are preserved under refrigerated conditions.

The limited number of samples as well as the composition of the surrounding plastic material and conditions of the HHP treatment might limit the generalizability of the findings.

Future work should be directed towards the traceability of other possible treatments and technological processing conditions that provide added value to the product, as well as considering other types of packaging, such as active packaging, modified atmosphere packaging, and cardboard-based packaging. This would contribute to the generation of knowledge for the control of traceability from an integral point of view in the Iberian sector.

## Figures and Tables

**Figure 1 foods-14-00432-f001:**
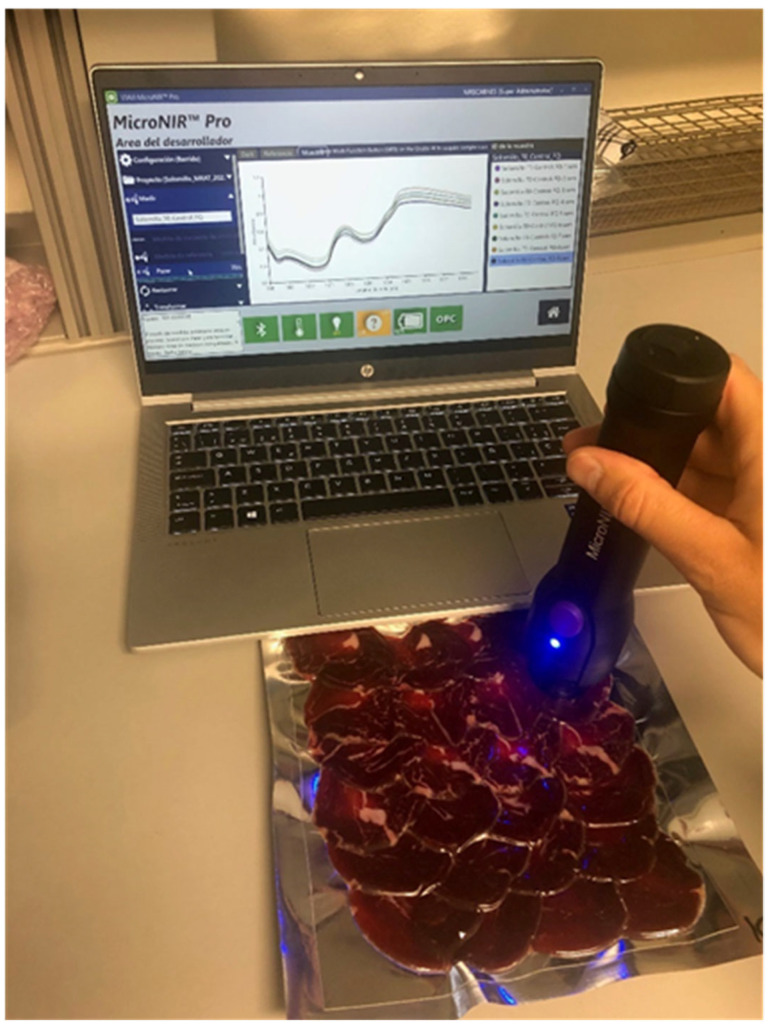
Spectra acquisition in Iberian dry-cured tenderloin without opening the package using the MicroNIR^TM^ 1700 OnSite-W (VIAVI) instrument.

**Figure 2 foods-14-00432-f002:**
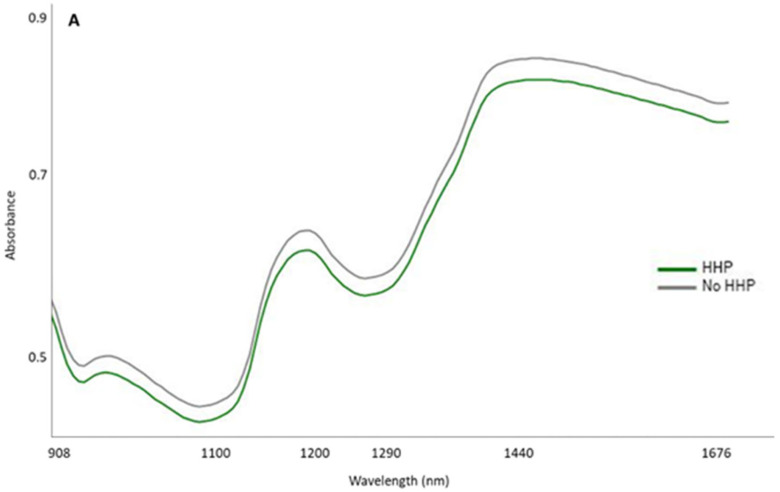
Raw mean spectra (absorbance) (908.1–1676.5 nm) of Iberian dry-cured tenderloin grouped according to the high hydrostatic pressure (HHP) treatment (**A**) and the preservation temperature over the course of 8 months (**B**).

**Figure 3 foods-14-00432-f003:**
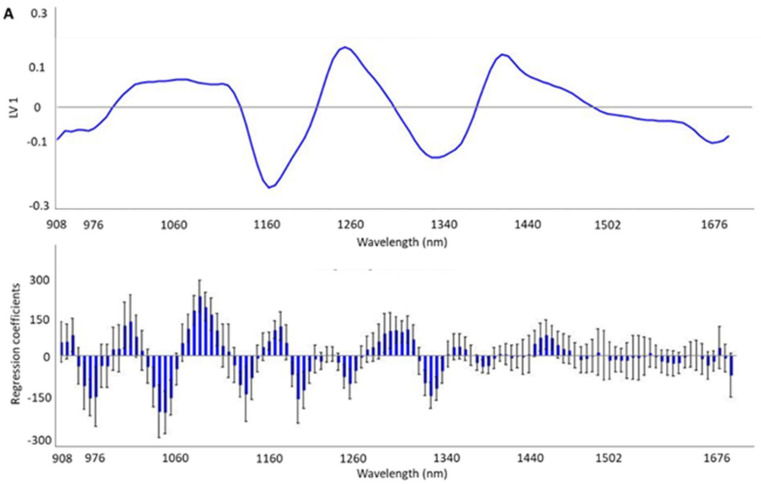
Loadings on the first LV and weighted regression coefficients in the projection of the best-fitting PLS-DA models developed by means of MicroNIR^TM^ 1700 OnSite-W (VIAVI) instrument to trace the high hydrostatic pressure (HHP) (**A**) and refrigeration conditions (**B**) in Iberian dry-cured *tenderloin* samples.

**Table 1 foods-14-00432-t001:** Distribution of samples used for calibration and validation sets according to the high hydrostatic pressure (HHP) treatment and preservation temperature.

Time (Months)	Pressurization Treatment	Temperature	Calibration	External Validation	Total
0	HHP	4 °C/20 °C	21	9	30
	No HHP	4 °C/20 °C	20	8	28
4	HHP	4 °C	21	9	30
		20 °C	21	9	30
	No HHP	4 °C	21	9	30
		20 °C	21	9	30
8	HHP	4 °C	21	9	30
		20 °C	21	9	30
	No HHP	4 °C	21	9	30
		20 °C	21	9	30
Total			209	89	298

HHP = high hydrostatic pressure; 0, 4, and 8 months of preservation in refrigeration (4 °C ± 2) or room-temperature (20 °C ± 2) conditions.

**Table 2 foods-14-00432-t002:** Statistics of the best-fitting PLS-DA model developed by means of MicroNIR^TM^ 1700 OnSite-W (VIAVI) instrument to discriminate the high hydrostatic pressure-treated Iberian dry-cured *tenderloin* samples from control ones.

Treatment	Pre-Treatment	Range (nm)	LVs	Cross-Validation			External Validation			
				N	1-VR	RMSECV	N	SE	SP	Accuracy
HHP	SNV + DE + SG 1,4,4,1	908.1–1676.5	16	161	0.78	0.233	89	60.00	61.36	60.67

HHP = high hydrostatic pressure; SNV = Standard normal variate; DE = de-trending; SG = Savitzky-Golay derivative, with the first number corresponding to the order derivative, the second and third ones indicating the smoothing points on the left and right sides and the last number corresponding to the polynomial; LVs = latent variables; n = number of samples; 1-VR = coefficient of determination in cross-validation; RMSECV = root mean square error of calibration; SE = sensitivity (%); SP = specificity (%).

**Table 3 foods-14-00432-t003:** Statistics of the best-fitting PLS-DA model developed by means of MicroNIR^TM^ 1700 OnSite-W (VIAVI) instrument to discriminate the Iberian dry-cured tenderloin samples preserved at refrigeration temperature from those preserved at room temperature over the course of 8 months.

Storage Temperature	Pre-treatment	Range (nm)	LVs	Cross-Validation			External Validation			
				n	1-VR	RMSECV	N	SE	SP	Accuracy
4 °C	SNV + DE + SG 1,4,4,1	908.1–1676.5	7	149	0.94	0.122	72	97.22	100.00	98.61

HHP = high hydrostatic pressure; SNV = standard normal variate; DE = de-trending; SG = Savitzky-Golay derivative, with the first number corresponding to order derivative, the second and third ones indicating the smoothing points on the left and right sides and the last number corresponding to the polynomial; LVs = latent variables; n = number of samples; 1-VR = coefficient of determination in cross-validation; RMSECV = root mean square error of calibration; SE = sensitivity (%); SP = specificity (%).

**Table 4 foods-14-00432-t004:** Influence of the high hydrostatic pressure (HHP) treatment on antioxidant and lipid profile and oxidative status of Iberian dry-cured tenderloin.

Treatment	Antioxidants		Main Fatty Acids Groups (g/100 g FAMEs)			Oxidative Status	
	α Tocoferol ug/g	ɣ Tocoferol ug/g	SFA	MUFA	PUFA	µg MDA/g	nmol Carbonyls/mg Protein
Control	12.25 ± 0.21	1.47 ± 0.04	39.02 ± 0.14	54.31 ± 0.18	6.53 ± 0.14	1.37 ± 0.04	3.49 ± 0.08
HHP	12.70 ± 0.22	1.40 ± 0.02	38.81 ± 0.13	54.57 ± 0.15	6.46 ± 0.13	1.52 ± 0.03	3.91 ± 0.11
*p* value	0.147	0.111	0.283	0.279	0.739	0.004	0.003

HHP = high hydrostatic pressure. Values are expressed as means ± standard error. FAMEs: fatty acid methyl esters. SFA: sum of the main saturated fatty acids detected (C12:0; C14:0, C16:0, C17:0; C18:0, C20:0); MUFA: sum of the main monounsaturated fatty acids detected (C16:1, C17:1, C18:1 n-9, C20:1 n-11); PUFA: sum of the main polyunsaturated fatty acids detected (C18:2 n-6, C18:3 n-3); MDA = malondialdehyde. Statistical significance was set at *p* ≤ 0.05.

**Table 5 foods-14-00432-t005:** Influence of storage temperature over the course of 8 months on antioxidant and lipid profile and oxidative status of Iberian dry-cured tenderloin.

Storage Temperature	Antioxidants		Main Fatty Acids Groups (g/100 g FAMEs)			Oxidative Status	
	α Tocoferol ug/g	ɣ Tocoferol ug/g	SFA	MUFA	PUFA	µg MDA/g	nmol Carbonyls/mg Protein
4 °C	12.05 ± 0.15	1.44 ± 0.03	38.67 ± 0.15	54.98 ± 0.13	6.30 ± 0.08	1.30 ± 0.03	3.84 ± 0.07
20 °C	11.18 ± 0.18	1.28 ± 0.03	39.37 ± 0.16	54.50 ± 0.13	5.93 ± 0.09	1.41 ± 0.03	3.68 ± 0.08
*p* value	0.000	0.000	0.002	0.013	0.003	0.026	0.117

Values are expressed as means ± standard error. FAMEs: fatty acid methyl esters. SFA: sum of the main saturated fatty acids detected (C12:0; C14:0, C16:0, C17:0; C18:0, C20:0); MUFA: sum of the main monounsaturated fatty acids detected (C16:1, C17:1, C18:1 n-9, C20:1 n-11); PUFA: sum of the main polyunsaturated fatty acids detected (C18:2 n-6, C18:3 n-3); MDA = malondialdehyde. Statistical significance was set at *p* ≤ 0.05.

## Data Availability

The original contributions presented in the study are included in the article/Appendix A, further inquiries can be directed to the corresponding author.

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
