# Peer review of "Near-Infrared Spectroscopy as a Tool for the Traceability Control of High-Quality Iberian Dry-Cured Meat Products"

_foods, 2025, doi:10.3390/foods14030432_

Round 1

Reviewer 1 Report

Comments and Suggestions for Authors

The present paper is dealing with efficiency of using NIRS as a tool to evaluate the quality of vacuum packed Iberian dry cured meat products. In general the paper is dealing with an important topic and is written well and is readable.

The abstract is well written and reflect the content of the study.

The introduction of the revised paper is OK and contains the main required references.

The materials and methods section is good.

The revised result and the discussion sections are OK

I have some specific comments and suggestions: 

Line 14-18 in the abstract has to be rewrite to improve the ease of understanding of the readers

Line 67 - in situ has to be in italic 

I may suggest that the authors will carefully go over the text and check and correct any other small errors.

I may recommend minor corrections and revisions.  

Reviewer 2 Report

Comments and Suggestions for Authors

The manuscript aims to explore of evaluate the potential of NIRS technology. To trace HHP treatment and storage temperature (4℃ vs. 20℃) on vacuum-packaged Iberian dry-cured tenderloin in a non-destructive way. Overall, NIRS could help to support the traceability of treatments that represent a high added value to the product. Authors are encouraged to consider the revisions raised below. 

1)It is recommended that the purpose and significance of the experiment be described in the introduction.

2)What is the specific processing time of 4℃ and 14℃ respectively? ï¼ˆline 97)

3)please type p in italic(lines 213,327, 345 and so on)

4)please add a margin of error in fig.1

5)Please briefly describe the changes in tocopherol, fat acid and oxidation status in the results. Whether these physical and chemical indicators and infrared spectrum correlation analysis.

6)Please add the limitations of this study.

Reviewer 3 Report

Comments and Suggestions for Authors

Translator        

This research utilizes near infrared spectroscopy combined with PLS-DA model to effectively trace the high hydrostatic pressure processing and preservation temperature over the course of a long-term in vacuum-packaged Iberian dry-cured tenderloin. The spectral analysis without opening the package is interesting and meaningful. However, the innovation is not enough. The discriminant method is based on a simple binary classification model. In view of the unique properties of vacuum-packaged Iberian dry-cured tenderloin, it is necessary to propose a novel traceability control strategy. In addition, the problem or challenge is not very well articulated in the introduction. The applications of NIRS for detection the quality of vacuum-packaged meat products need to be further introduced.

1. The problem or challenge is not very well articulated in the introduction. The applications of NIRS in the quality detection of vacuum-packaged meat products needs to be further introduced.

2. It is recommended to consider a multitasking model for synchronous traceability. PLS-DA is a typical linear model, and nonlinear models should also be used to compare model performance.

3. Plastic affects the penetration of light. How do the authors demonstrate that the reflection correction eliminates the spectral interference of vacuum packaging plastics?

4. A discussion should provide a critical view on own findings by comparing own results with results from other authors. Spectral interpretation is insufficient to reflect the innovation of the method.

5. Future research directions should be explained.

Future research directions should also be explained.

Reviewer 4 Report

Comments and Suggestions for Authors

This manuscript applied near-infrared spectroscopy to identify Iberian dry-cured meat products,  tracing the high hydrostatic pressure processing and preservation temperature. There are the following questions:

1. Introduction:  the research purpose of this paper is not very prominent, and some additional content is needed,  such as the application of HHP.

2. In the part of Materials and Methods, the description of experimental methods is ok. However,  how you assign the samples, only 80 samples used, but inTable 1, it's different.

3.The number of samples is limited. The study  raises questions about the generalizability of the findings. The authors  fail to address whether the proposed methods can be applied to other conditions.

3.Why you choose 4ºC/20ºC, how you control the 20ºC?

4. Results mainly lacked in data presentation in section of 3.3 and 3.4.

5. Conclusions shoud contain the main findings and results, please modify it.

The overall feeling is that the workload and innovation of the paper are not enough

Comments on the Quality of English Language

Check the English expression, punctuation, especially commas. 

Round 2

Reviewer 3 Report

Comments and Suggestions for Authors

All of my previous concerns have been addressed properly. I do not have further concern.

Author Response

Comments and Suggestions for Authors "All of my previous concerns have been addressed properly. I do not have further concern"

Response: The authors again thank the reviewer for his time and dedication in reviewing this manuscript.

Reviewer 4 Report

Comments and Suggestions for Authors

The authors have made modifications and answered the questions. But question: Results mainly lacked in data presentation or visualization.

Why you add analyze the Effect of the HHP processing and preservation temperature on the α- and γ-tocopherol, fatty acid profil and oxidative status? What is the relationship of your conclusions. No results images presented.

The number of samples is not enough, and the workload and innovation is lacked although the authors explained..

I think it's not a well modified manuscript.

Comments on the Quality of English Language

Check the English expression, punctuation, especially commas. 

Author Response

Comments and Suggestions for Authors “Why you add analyze the Effect of the HHP processing and preservation temperature on the α- and γ-tocopherol, fatty acid profil and oxidative status? What is the relationship of your conclusions. No results images presented.”

Response: The authors now understand the reviewer's comment and will try to respond as well as possible.

The effect of the HHP processing and preservation temperature on the tocopherol content, fatty acid profile and oxidative status were added to the current research study because the authors consider that the possibility of tracing both factors by means of NIRS technology depends to a large extent on these parameters. Therefore, the study of both factors on these parameters would help to explain the results obtained. Indeed, absorption bands of the NIR spectra are characterized by specific functional chemical groups, and the ability to trace the HHP treatment as well as the preservation temperature may be attributed to absorbance differences.  And as detailed extensively in the discussion, the spectral differences between the spectra of the samples from the different groups are related to tocopherol content, fatty acid profil and oxidative status.

However, as the objective of the present manuscript was to evaluate the potential of NIRS technology to trace HHP treatment and storage temperature on vacuum-packaged Iberian dry-cured tenderloin, the authors preferred to incorporate the effect of these factors at the physical-chemical level into the supplementary material.

Comments and Suggestions for Authors “The number of samples is not enough, and the workload and innovation is lacked although the authors explained…I think it's not a well modified manuscript.”

Response: These issues were properly defended in the previous review. The authors have nothing further to add.